# THINK IN PARALLEL, ANSWER AS ONE: LOGIT AVERAGING FOR OPEN-ENDED REASONING

**Haonan Wang**[*1]  **Chao Du**[2]  **Kenji Kawaguchi**[1]  **Tianyu Pang**[†2]
[1]National University of Singapore  [2]Sea AI Lab, Singapore
haonan.wang@u.nus.edu;  kenji@comp.nus.edu.sg;
{tianyupang, duchao}@sea.com

## ABSTRACT

Majority voting has proven effective for close-ended question answering by aggregating parallel reasoning traces. However, it is not directly applicable to *open-ended* reasoning, such as code generation and web-based deep research, where a "majority" over complete solutions is ill-defined. We introduce THINKMERGE, a *training-free*, *plug-and-play* decoding strategy that runs $K$ parallel reasoning traces and averages their next-token *logits* at synchronization points to produce a single coherent output. THINKMERGE integrates seamlessly with vLLM/SGLang and remains compatible with standard decoding techniques such as Top-$p$/Top-$k$. Empirically, it matches or surpasses majority voting on AIME and GPQA, while delivering consistent gains on open-ended coding tasks: on LiveCodeBench (hard), pass@1 improves by **+8.28%** for DeepCoder-14B-Preview and **+7.58%** for Qwen3-8B. Beyond code, we further show that THINKMERGE improves web-based deep-research agents (e.g., WebSailor-7B/32B) across GAIA, BrowseComp-en/zh, and XbenchDeepSearch. These results demonstrate that parallel test-time scaling can benefit open-ended reasoning without relying on voting over complete outputs.

## 1 INTRODUCTION

Recent advances in Large Language Models (LLMs) have been driven by test-time compute scaling. As evidenced by OpenAI's o1 (OpenAI, 2024), DeepSeek-R1 (Guo et al., 2025), etc., models generate extended "think" segments that reflect intermediate hypotheses, derivations, and self-corrections prior to emitting the final answer (Chen et al., 2025b; Yang et al., 2025c). Such *sequential* test-time scaling has established a new paradigm: increasing the inference-time computation (e.g., longer reasoning traces) often leads to improved accuracy and problem-solving capability.

Yet simply lengthening the chain has diminishing returns and can even hurt, e.g., overthinking (Chen et al., 2024; Cuadron et al., 2025), with studies showing that correct answers often appear in shorter traces (Zeng et al., 2025). A natural complement is *parallel* scaling: generating multiple reasoning traces and combining their evidence, most effectively through majority voting on close-ended tasks (Wang et al., 2022; Aggarwal et al., 2023; Brown et al., 2024; Knappe et al., 2024).

Many real-world workloads, however, are inherently *open-ended*. Coding assistants must output executable programs (Jimenez et al., 2024; Yang et al., 2025b), while autonomous deep research agents often need model context protocol (MCP) tool calling, multi-step plans and long-form explanations (OpenAI, 2021; Anthropic, 2025; Alibaba-NLP, 2025). In such settings, majority voting is undefined since there is no single canonical answer, even though it has been highly effective in math and QA (MAA, 2025; Cobbe et al., 2021; Rein et al., 2024). This gap motivates a central question: *Can the benefits of test-time parallel reasoning be extended to open-ended tasks without relying on voting over complete outputs?*

In this work, we address this question by introducing THINKMERGE, an inference-time framework that averages logits across parallel reasoning paths to construct a single high-quality answer. Unlike

---

[*]Work done during Haonan Wang's internships at Sea AI Lab.
[†]Correspondence to Tianyu Pang.

Figure 1: Margins between *Pass@1* and *Pass@8* across (close-ended) tasks and models. Larger margins imply more room for majority voting to help.

majority voting, which selects among complete outputs, THINKMERGE enables the model to think in parallel but speak with one voice. Concretely, given a question, we run $K$ diverse reasoning traces concurrently. At a synchronization point (for example, after a reasoning delimiter), we aggregate the next-token logits from all traces by averaging them, normalize the merged logits into a probability distribution, and sample the next token. The chosen token is then injected back into every trace, as if each had generated it, and the parallel reasoning continues step by step.

Through this iterative ensemble decoding, the model produces a single coherent solution that reflects the guidance of multiple concurrent "*thoughts*". The approach is entirely training-free: it requires no fine-tuning or additional supervision, only multiple forward passes during inference. Moreover, it is plug-and-play and fully compatible with standard decoding strategies such as Top-$p$, Top-$k$, temperature, and repetition penalties (Shi et al., 2024). Intuitively, THINKMERGE allows the model to explore a broader range of ideas in parallel and converge on a more reliable answer.

Empirically, we evaluate THINKMERGE on both closed-ended and open-ended reasoning tasks. On math and science benchmarks with well-defined answers, such as AIME (MAA, 2025) and GPQA (Rein et al., 2024), THINKMERGE matches or slightly surpasses the accuracy of majority voting and its variants (Wang et al., 2022; Zeng et al., 2025). More critically, on *open-ended* tasks where majority voting is not applicable, THINKMERGE yields consistent improvements over single-chain decoding. For example, on LiveCodeBench (Jain et al., 2024), Pass@1 increases by **+8.28%** for DeepCoder-14B-Preview and **+7.58%** for Qwen3-8B on the hard-level coding problems. Beyond coding, we further demonstrate that THINKMERGE also benefits web-based deep-research agents: running multiple WebSailor-7B/32B (Li et al., 2025a) trajectories in parallel and merging their logits in decoding answers improves pass@1 by up to **+10.2%** points on XbenchDeepSearch (Chen et al., 2025a) and **+4.8%** points on GAIA (Mialon et al., 2023). THINKMERGE integrates seamlessly with inference frameworks such as vLLM (Kwon et al., 2023) and SGLang (Zheng et al., 2024), supports both online serving and offline batch decoding, and remains compatible with standard sampling controls (Top-$p$, Top-$k$, temperature, penalties). It can thus be adopted as a simple drop-in augmentation to existing LLM deployments.

## 2 RELATED WORK

**Majority Voting and Variants.** Parallel scaling explores *many* candidate solutions and aggregates them (Brown et al., 2024; Zeng et al., 2025; Stroebl et al., 2024; Sun et al., 2024; Gui et al., 2024; Snell et al., 2025; Liu et al., 2025; Wu et al., 2025a; Jiang et al., 2023; Li et al., 2025c; Chen et al., 2023). Aggregation can happen at the *solution level*, either with reward-guided Best-of-$N$ search (Sun et al., 2024) or guidance-free voting such as rule-based Majority Voting (Wang et al., 2022; Chen et al., 2023), with variants that adapt the sample count or filter candidates (Aggarwal et al., 2023; Xue et al., 2023; Huang et al., 2024; Knappe et al., 2024). While these methods deliver strong gains on *closed-ended* tasks, they are ill-defined for *open-ended* reasoning, where valid outputs rarely repeat and "voting" is not meaningful. Instead, we ensemble logit-level (pre-softmax) during the answer generation phase across reasoning paths, reducing dependence on a single consensus answer and turning extra test-time computation into performance gains on open-ended tasks.

**Model-Based Aggregation.** Beyond voting, several model-based aggregation methods have been proposed (Chen et al., 2023; Qi et al., 2025; Zhao et al., 2025; Jiang et al., 2023; Edge et al., 2024). These either (i) train a separate scorer to select among candidates, or (ii) prompt an LLM to compare

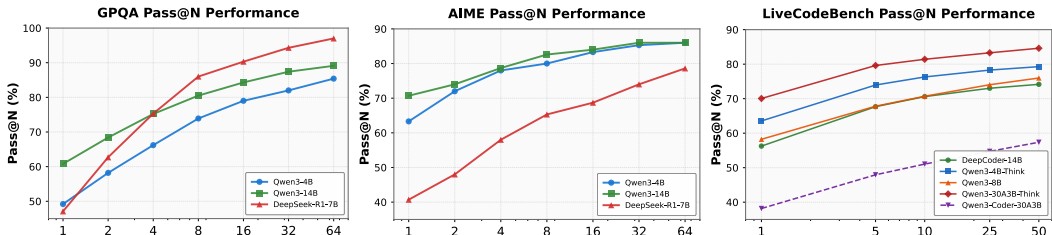

Figure 2: Trend of *Pass@N* as the number of samples $N$ increases. Gains are evident on AIME 2025, GPQA Diamond, and the open-ended LIVECODEBENCH.

and summarize them. For example, LLM-Blender (Jiang et al., 2023) takes the top-$K$ candidate answers of a query, concatenates them with the input, and feeds the resulting sequence into a model that generates a new answer intended to outperform all individual candidates. GraphRAG (Edge et al., 2024) adopts a similar high-level map-reduce pattern on the retrieval side: it first produces partial summaries and answers in parallel, then issues a single LLM call to *summarize* these partial answers into a final response. Such approaches require an additional model to assign scalar scores or produce summaries, (in training) typically demand supervised tuning or domain adaptation to learn reliable judgments, and (in inference) incur at least one extra model forward over long, concatenated candidate outputs. In contrast, our method is *training-free* and performs *logit-level* aggregation during answer decoding, yielding a single output that integrates multiple parallel reasoning paths.

**Probability-Level and PoE-Style Ensembles.** A separate line of work ensembles next-token *probabilities* at each decoding step. Classical Product-of-Experts (PoE) (Hinton, 1999) combines expert predictions by multiplying their probability distributions. Recent LLM work adapts this idea in probability space. M-Ped (Guo et al., 2024) submits multiple prompt variants for the same input in batch mode and ensembles the next-token distribution by averaging the per-prompt probabilities. Other approaches ensemble different models: EVA (Xu et al., 2024) learns cross-model vocabulary mappings to align output distributions into a shared space before averaging, while the agreement-based method of Wicks et al. (Wicks et al., 2025) constrains multiple models with different vocabularies to generate a shared surface string via an efficient search procedure. These probability-level token ensembles primarily target settings with multiple prompts or multiple models, usually on closed-form generation tasks such as machine translation or data-to-text. THINKMERGE is related in spirit but differs in how and where aggregation is applied. We operate within a *single* reasoning model in the modern "think–then–answer" paradigm, treat $K$ chain-of-thought traces as experts, and aggregate their *pre-softmax logits* only during the answer phase, leaving the thinking phase fully diverse. This logit-space fusion can be viewed as a PoE-style combination implemented as a drop-in logit processor in standard decoding framworks (e.g., vLLM), and, more importantly, allows us to systematically convert extra test-time compute into accuracy and success-rate gains on *open-ended* reasoning and agentic deep-research benchmarks where solution-level voting is not directly applicable.

## 3 PRELIMINARY STUDY

To study the relation between ensemble gains and *answer coverage*—the probability that among $K$ sampled solutions at least one is correct (i.e., Pass@$K$), we evaluate closed-ended math/science (AIME'24/'25 (MAA, 2025), HMMT'24/'25 (Balunović et al., 2025), GPQA (Rein et al., 2024)) and open-ended coding (LiveCodeBench v5 (Jain et al., 2024)). For closed-ended tasks we use Qwen3-1.7B/4B/8B/14B (Yang et al., 2025a) and DeepSeek-R1-Distill-Qwen-7B (Guo et al., 2025); for coding we test DeepCoder-14B-Preview (Luo et al., 2025), Qwen3-8B, Qwen3-Coder-30A3B-Instruct, Qwen3-4B-Thinking, Qwen3-Think-30A3B (Yang et al., 2025a).

**Closed-ended tasks: Majority@$K$ and Pass@$K$.** For multiple-sampling at inference time, the empirical benefit of *majority voting* on closed-ended benchmarks is closely tied to the improved margins between Pass@$K$ and Pass@1 and how quickly *Pass@$K$* grows with $K$. Intuitively, when additional samples quickly increase the probability that the correct option appears (i.e., larger pass@$K$ gaps between $K{=}1$ and $K{>}1$), the vote distribution shifts toward the right answer; when pass@$K$ saturates, samples tend to reinforce the same wrong choice and voting yields little gain. As shown in Figure 1, there is a clear margin between *Pass@8* and *Pass@1* for two reasoning models across four closed-ended datasets, indicating that parallel sampling meaningfully raises the chance of observing the correct option—hence majority voting is expected to perform between these bounds.

**Does parallel sampling help in open-ended settings?** Unlike classification, open-ended problems do not admit a direct vote over a small, discrete label set. We therefore examine whether the *existence*

Figure 3: LIVECODEBENCH stratified by difficulty. Hard questions exhibit larger *Pass@N* gains as $N$ increases, indicating that parallel reasoning is potentially helpful on challenging instances.

signal captured by pass@N (at least one good solution among $N$ samples) still grows with $N$ when evaluation is based on program execution or unit tests. Figure 2 plots *Pass@N* as $N$ increases. We observe a rapid rise on closed-ended AIME 2025 and GPQA Diamond, and a consistent increase on the open-ended LIVECODEBENCH. The positive slope on LIVECODEBENCH indicates that ensembling multiple reasoning trajectories can be potentially beneficial.

**Where do the gains come from?** To localize the effect, we stratify LIVECODEBENCH by difficulty. Figure 3 shows that the increase in *Pass@N* is more pronounced on the hard subset: difficult problems benefit more from multiple, diverse reasoning attempts. This pattern mirrors closed-ended observations—hard items accrue larger returns from additional samples—suggesting that *ensembling over parallel thoughts* might unlock solutions that single-pass decoding misses on hard problems.

**Takeaways and motivation.** Across closed- and open-ended settings, *Pass@N* improves with $N$, and the gains concentrate on harder instances. For closed-ended tasks, majority voting directly converts these gains into accuracy. For open-ended tasks, however, voting over free-form outputs is ill-posed. These findings motivate an *open-ended* ensembling mechanism that aggregates thinking processes—precisely the goal of our approach that averages token-level logits across parallel reasoning paths.

## 4 METHOD

We propose a training-free, plug-and-play decoding strategy that ensembles *diverse chains of thought* at the token level to produce a single coherent answer for open-ended queries. The method proceeds in two stages: (i) generate $K$ diverse reasoning traces up to a delimiter token, e.g. `</think>`; (ii) *after* the delimiter, decode one shared answer sequence by averaging the next-token *logits* across all $K$ reasoning contexts at every autoregressive step.

**Diverse Reasoning Generation.** Given an input prompt or question $Q$, we prompt the LLM to produce a step-by-step reasoning ending with a special delimiter marker (e.g. `</think>` to indicate the end of the thinking segment). To obtain diverse reasoning traces, we sample $K$ independent chain-of-thought sequences $R_1, R_2, \ldots, R_K$ from the model. Notably, official model cards for recent reasoning models recommend relatively high temperatures (e.g., 0.5–0.7 for DeepSeek and Qwen) (Guo et al., 2025; Yang et al., 2025a). Because of the randomness introduced by the high temperature, the $K$ reasoning paths are varied, exploring different plausible approaches or perspectives to the problem. This step uses the model as-is, without any fine-tuning – we are simply drawing multiple reasoning samples from the model's own distribution, which makes the procedure straightforward to integrate.

**Ensembled Answer Decoding.** Once desired number of reasoning chains reach delimiter markers (i.e. the end of the thought process), the model begins generating the answer portion jointly informed by all chains. At each autoregressive decoding step $i$ of the answer, we query the model's next-token *pre-softmax logits* $M_\theta(\cdot)$ for each reasoning chain context. We then aggregate these logits by arithmetic mean. Formally, let $y_{<1}$ denote an empty answer prefix, for each chain $k \in \{1, \ldots, K\}$, we define the logit vector over the vocabulary $\mathcal{V}$ as $\mathbf{z}_i^{(k)} = M_\theta(Q, R_k, y_{<i}) \in \mathbb{R}^{|\mathcal{V}|}$, we then ensemble *on logits* via arithmetic mean and only then apply softmax:

$$\bar{\mathbf{z}}_i = \frac{1}{K} \sum_{k=1}^{K} \mathbf{z}_i^{(k)}, \qquad \bar{P}_\theta(y_i \mid Q, R_{1..K}, y_{<i}) = \text{softmax}(\bar{\mathbf{z}}_i)[y_i].$$

for each possible token $y_i \in \mathcal{V}$ at that step. We then sample or select the next token $y_i$ from this aggregated distribution $\bar{P}$. Note, this ensemble step will not have impact on the up-following

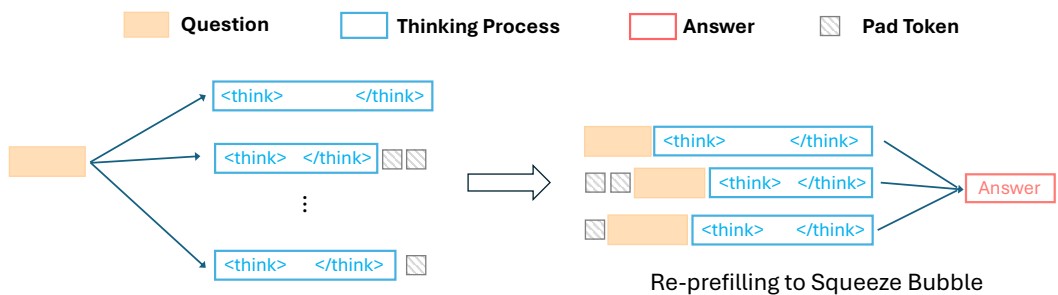

Figure 4: **Two-stage implementation.** *Stage I - Map (Diverse reasoning).* Sample $K$ chain-of-thought traces up to a delimiter. *Stage II - Reduce (Answer decoding).* Left-pad the *question+reasoning* contexts to a common length, re-prefill to *"squeeze bubble"*, and then decode a single answer sequence by averaging the pre-softmax logits across the $K$ reasonings at every step.

---

**Algorithm 1** Ensemble-of-Thought

---

**Require:** LLM $M_\theta$ (outputs pre-softmax logits); query $Q$; number of traces $K$; reasoning temperature $\tau_{\text{think}}$; answer temperature $\tau_{\text{ans}}$; decoding policy $\pi$ (e.g., greedy / top-$k$ / top-$p$ / repetition-penalty); stopping rule STOP (eos/length/validator)

1: **Parallel thinking:** For $k = 1, \ldots, K$, sample a reasoning trace $R_k \sim p_\theta(\cdot \mid Q; \tau_{\text{think}})$ by running the model until reasoning end delimiter.
2: Initialize the shared answer prefix $y \leftarrow \varnothing$.
3: **while not** STOP$(y)$ **do**
4:     **for** $k = 1$ to $K$ **do**                                     ▷ fully parallelizable across $k$
5:         $\ell^{(k)} \leftarrow M_\theta(Q, R_k, y)$             ▷ next-token *logits* conditioned on $(Q, R_k, y)$
6:     **end for**
7:     $\bar{\ell} \leftarrow \frac{1}{K} \sum_{k=1}^{K} \ell^{(k)}$
8:     *(optional)* Apply logit policy $\pi$ on the *averaged* logits: $\bar{\ell} \leftarrow \text{PROCESS}(\bar{\ell}; \pi)$
9:     Form the answer-step distribution $\bar{P} \leftarrow \text{softmax}(\bar{\ell}/\tau_{\text{ans}})$
10:    Select the next token $y_{\text{next}} \sim \pi(\bar{P})$     ▷ greedy: $\arg\max$; sampling: draw from $\bar{P}$
11:    $y \leftarrow y \parallel y_{\text{next}}$                              ▷ append to the shared answer prefix
12:    *Note:* All $K$ contexts implicitly share the updated $y$ token at the next step via $M_\theta(Q, R_k, y)$.
13: **end while**
14: **return** $y$

---

decoding strategies, such as Top-k, temperature, and penalty (Shi et al., 2024). This chosen token $y_i$ becomes the next word in the final answer and is also appended to each of the $K$ contexts before proceeding to the next decoding step. By updating all contexts with the same generated answer token, we ensure that subsequent probability predictions from each chain remain conditioned on a common partial answer. We repeat this token-level ensemble process autoregressively until an end-of-answer token is produced or another stopping criterion is met.

## 4.1 IMPLEMENTATION

Our method integrates cleanly with modern high-throughput inference stacks, including vLLM (Kwon et al., 2023) and SGLang (Zheng et al., 2024).

**Two-stage pipeline.** As illustrated in Figure 4, we (i) *batch-generate* $K$ diverse reasoning traces up to a delimiter, and then (ii) *left-pad* all *question+reasoning* contexts to the same length and *re-prefill* to build an aligned KV cache. This "bubble squeezing" removes idle compute caused by unequal trace lengths and enables *logit-level ensembling* for the answer: at each autoregressive step we average the pre-softmax logits from the $K$ contexts and decode a single shared token. The design works in both online serving and offline batch settings for vLLM/SGlang with minimal changes. In practice, prefill in modern optimized systems is fast; its overhead is negligible compared to decoding, so the two-stage variant remains efficient while being easy to instrument for ablations studies.

**One-step pipeline with Flex-Attention.** Alternatively, we integrate ensembling directly into the decoding (Figure 5). We treat the $K$ sequences as a batch and rely on flexable masks of *Flex-Attention* (PyTorch Team, 2025) to *silence* padding tokens emitted by shorter traces when waiting the longest reasoning stream completes. After the delimiter, we aggregate the *pre-softmax logits*

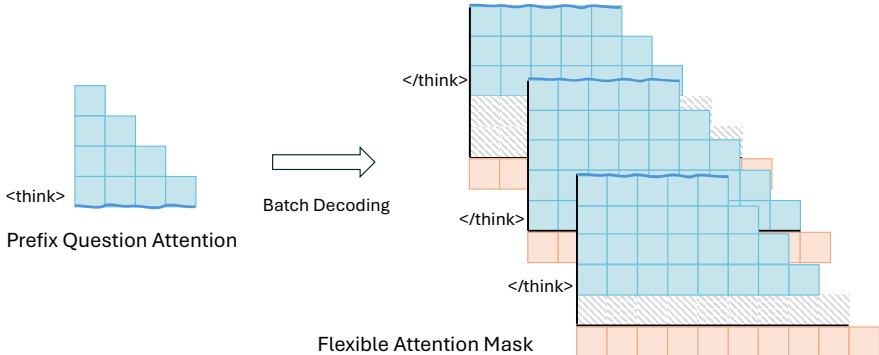

Figure 5: **One-step decoding with Flex-Attention.** Padding tokens produced by shorter reasoning traces when waiting for the longest trace are masked so they are not attended. After the delimiter (e.g., `</think>`), we average pre-softmax logits across all streams to produce a single shared answer token at each step for all streams.

across the $K$ contexts at every step and produce one shared answer token. We implement this variant in the HuggingFace Transformers generation pipeline (Wolf et al., 2020), leveraging its stable Flex-Attention support (Hugging Face, 2024). At the time of our experiments, vLLM was in the process of integrating Flex-Attention into vLLM v1 (drisspg, 2025); we plan to open-source a vLLM v1 implementation of the one-step variant once upstream support stabilizes.

For the controlled analyses and method variants in Section 4.2, we default to the *two-stage* vLLM pipeline, which maintains high throughput while providing convenient handling of reasoning traces.

## 4.2 DESIGN CHOICES AND VARIANTS

We study four orthogonal design reasoning trace processing startegies for *when* to start answer decoding and *which* reasoning traces to ensemble. Unless noted, we sample $K$ reasoning traces with temperature $\tau$, stop each at a delimiter token, and then ensemble a subset of them to decode one shared answer by averaging pre-softmax logits.

**(A) Direct-Merge.** We decode $K$ reasoning traces in parallel until their delimiter and immediately ensemble them to decode the answer. This is the default configuration used in most experiments. It can be regarded as no extra processing.

**(B) $K$ Early-Ready.** To reduce tail latency from very long traces, we begin answer decoding as soon as $K$ traces have completed their reasoning segments, rather than waiting for all $N(N > K)$ reasoning to be finished. Formally, let $\mathcal{R}_{\text{ready}} = \{R_k : R_k \text{ has emitted the delimiter}\}$. We start ensembling when $|\mathcal{R}_{\text{ready}}| \geq K$. The answer is then decoded by averaging logits over the currently available thinkings. This variant trades a small amount of diversity for lower latency and higher throughput, and is useful in online serving.

**(C) Trimming (De-Repeat Suffix).** Motivated by prior observations that models may emit repeated reflection fragments (e.g., *"Wait"*, *"Hmm"*, and *"Alternatively"*) near the end of the reasoning phase and that overthinking can degrade performance (Wang et al., 2025), we remove degenerate repeated suffixes before re-prefill. Concretely, for each finished reasoning trace $R_k$, we detect the longest repeated suffix (e.g., via regex pattern matching with length thresholds) and trim it, producing $\tilde{R}_k = \text{trim}(R_k)$. This preserves semantically useful steps while avoiding overweighting on the long and misleading words when decoding final answers.

**(D) Shortest-$K$ Merge (Anti-Overthinking).** Prior work reports that excessive "*overthinking*" can correlate with worse final answers (Chen et al., 2024; Cuadron et al., 2025; Wu et al., 2025b; Sui et al., 2025). To bias toward concise, high-signal reasoning, we sort completed traces by their pre-delimiter length and select the $K$ *shortest* from a reasoning pool with size $N(N \gg K)$ for logit ensembling: $\mathcal{S} = \text{argsort}\left(\{\text{len}(R_k)\}_{k=1}^{N}\right)_{1:K}$, ensemble over $\{R_k\}_{k \in \mathcal{S}}$. This variant leverages a length–quality inductive bias to stay clear and on topic, and they help avoid late drift or repetition. Different from $K$-Early-Ready, it waits for all $N$ traces to finish so the $K$ shortest can be selected globally, trading latency for an anti-overthinking bias.

## 5 EXPERIMENTS

**Experimental Setup.** We evaluate THINKMERGE in two regimes. (i) For *closed-ended* reasoning, we consider AIME 2025 (MAA, 2025) and GPQA Diamond (Rein et al., 2024), where each question

Table 1: Performance of Majority Voting (MV) vs. THINKMERGE on AIME'25 across different strategies. Within each (model $\times$ $K$ $\times$ strategy) group, the highest score within each category is **bold**; the highest within tier is underlined.

| Model | All-Reduce | | | Early-Ready | | Shortest-$K$ Merge | |
|---|---|---|---|---|---|---|---|
| | MV | DirectMerge (A) | Trimming (B) | MV | Ours (C) | MV | Ours (D) |
| $K = 2$ | | | | | | | |
| Qwen3-4B | 63.3 | **66.7** | 66.0 | 66.7 | 66.7 | **70.7** | 65.3 |
| Qwen3-14B | 70.7 | 72.0 | **72.7** | 72.0 | **73.3** | **75.3** | 74.7 |
| R1-Distill-Qwen-7B | 40.7 | 41.3 | **42.0** | 40.7 | **41.3** | **50.7** | 48.0 |
| $K = 4$ | | | | | | | |
| Qwen3-4B | 68.0 | **72.0** | 68.0 | 72.7 | 72.7 | **75.3** | 69.3 |
| Qwen3-14B | 73.3 | 72.0 | 73.3 | **76.0** | 73.3 | **78.0** | 76.0 |
| R1-Distill-Qwen-7B | **47.3** | 46.0 | 46.0 | **47.3** | 45.3 | **52.0** | 50.7 |
| $K = 8$ | | | | | | | |
| Qwen3-4B | 68.7 | **70.0** | 68.7 | **73.3** | 70.8 | **75.3** | 72.7 |
| Qwen3-14B | 74.0 | **78.0** | 73.3 | 77.4 | **78.0** | **80.0** | 78.7 |
| R1-Distill-Qwen-7B | 46.7 | 45.3 | **48.0** | 46.7 | 45.3 | 52.7 | 52.7 |

Table 2: Performance of Majority Voting (MV) and THINKMERGE on GPQA across different strategies. Within each (model $\times$ $K$ $\times$ strategy) group, the highest score within each category is **bold**; the highest within tier is underlined.

| Model | All-Reduce | | | Early-Ready | | Shortest-$K$ Merge | |
|---|---|---|---|---|---|---|---|
| | MV | DirectMerge (A) | Trimming (B) | MV | Ours (C) | MV | Ours (D) |
| $K = 2$ | | | | | | | |
| Qwen3-4B | 49.2 | **50.3** | 50.0 | 49.4 | **50.3** | 52.0 | **52.6** |
| Qwen3-14B | 60.8 | **61.6** | 61.4 | 60.8 | **62.1** | 63.4 | **63.7** |
| R1-Distill-Qwen-7B | 44.9 | 49.2 | 49.2 | 47.4 | **49.0** | **47.1** | 43.8 |
| $K = 4$ | | | | | | | |
| Qwen3-4B | 51.2 | 52.2 | 52.2 | 51.4 | **52.4** | **53.7** | 51.8 |
| Qwen3-14B | 63.0 | **64.0** | 62.7 | **64.0** | 62.4 | **65.2** | 63.8 |
| R1-Distill-Qwen-7B | **50.3** | 48.9 | 49.0 | **50.3** | 48.9 | **50.2** | 47.4 |
| $K = 8$ | | | | | | | |
| Qwen3-4B | **53.3** | 51.6 | 51.3 | **53.3** | 51.3 | **55.5** | 53.8 |
| Qwen3-14B | 63.9 | **64.1** | 63.0 | 64.1 | 64.1 | **65.9** | 63.7 |
| R1-Distill-Qwen-7B | **52.5** | 50.0 | 50.0 | **52.5** | 50.2 | **52.9** | 47.2 |

has a unique ground-truth answer and multiple samples can be aggregated via majority voting. (ii) For *open-ended* reasoning, we evaluate on LiveCodeBench v5 (2024.10–2025.02) (Jain et al., 2024), BrowseComp-en (Wei et al., 2025), BrowseComp-zh (Zhou et al., 2025), GAIA (Mialon et al., 2023), and xBench-DeepSearch (Chen et al., 2025a), where majority voting is ill-defined and the quality of a single coherent solution is what matters. For closed-ended tasks, we run five trials and report the mean accuracy in the main text, with mean±std in Appendix A.2; for LiveCodeBench, we report pass@1.

For each question, we generate $K \in \{2, 4, 8\}$ parallel reasoning traces and perform an ensemble step at the answer phase by averaging pre-softmax logits across the $K$. For the *Shortest-$K$ Merge* variant (anti-overthinking), we first produce a pool of $N = 64$ completed traces and ensemble the $K$ shortest by pre-delimiter length. Our processing strategies, (B) *Early-Ready* and (D) *Shortest-$K$ Merge*, can also be paired with majority voting (MV). For fairness, we report MV under the same operation whenever it is applicable; thus, trimming cannot be applied to it. Methods that aggregate *all* completed traces—*Majority Voting*, *Direct-Merge*, and *De-Repeat Suffix Trimming*—are grouped under the label **All-Reduce** in the tables. In contrast, **Early-Ready** and **Shortest-$K$** operate on a subset of traces during merging, so we place them in separate columns.

For closed-ended tasks we evaluate Qwen3-4B, Qwen3-14B (Yang et al., 2025a), and DeepSeek-R1-Distill-Qwen-7B (Guo et al., 2025). For the open-ended coding task we use DeepCoder-14B-Preview (Luo et al., 2025), Qwen3-8B, Qwen3-Coder-30A3B-Instruct, Qwen3-4B-Thinking (0725), and Qwen3-Think-30A3B (Yang et al., 2025a). For open-ended deep-research agent task, we use the WebSailor-3B, WebSailor-7B and WebSailor-32B (Li et al., 2025a). We set the maximum sequence length to $32{,}768$, detailed sampling hyperparameters in Appendix A.1.

Table 3: Effect of answer-phase temperature for THINKMERGE. Default vs. setting the answer-phase temperature to $T_{\mathrm{ans}}{=}0.3$. The highest score within each category is **bold**; the tier is underlined.

| Model | Direct-Merge | | Early-Ready | | Shortest-$K$ Merge | |
|---|---|---|---|---|---|---|
| | Default | $T_{\mathrm{ans}}{=}0.3$ | Default | $T_{\mathrm{ans}}{=}0.3$ | Default | $T_{\mathrm{ans}}{=}0.3$ |
| $K=2$ | | | | | | |
| Qwen3-4B | **66.7** | 64.7 | 66.7 | **67.3** | 65.3 | **66.7** |
| Qwen3-14B | **72.0** | 71.3 | 73.3 | 73.3 | **74.7** | 74.0 |
| R1-Distill-Qwen-7B | 41.3 | 41.3 | 41.3 | 41.3 | 48.0 | **48.7** |
| $K=4$ | | | | | | |
| Qwen3-4B | **72.0** | 70.0 | **72.7** | 72.0 | 69.3 | **70.7** |
| Qwen3-14B | 72.0 | **72.7** | **73.3** | 72.7 | **76.0** | 75.3 |
| R1-Distill-Qwen-7B | **46.0** | 42.7 | **45.3** | 43.3 | 50.7 | **52.0** |
| $K=8$ | | | | | | |
| Qwen3-4B | **70.0** | 68.7 | **70.8** | 69.3 | **72.7** | 70.7 |
| Qwen3-14B | **78.0** | 74.7 | **78.0** | 76.7 | **78.7** | 77.3 |
| R1-Distill-Qwen-7B | **45.3** | 44.7 | **45.3** | 44.0 | **52.7** | 51.3 |

Table 4: LiveCodeBench Overall Pass@1 (%). Row-wise best among merge settings is highlighted.

| Model | Baseline | Direct-Merge | | | Shortest-$K$ Merge | | |
|---|---|---|---|---|---|---|---|
| | | $K{=}8$ | $K{=}4$ | $K{=}2$ | $K{=}8$ | $K{=}4$ | $K{=}2$ |
| DeepCoder-14B-Preview | 55.32 | 56.23 | 57.14 | 58.36 | 59.57 | 59.88 | **61.09** |
| Qwen3-8B | 57.14 | 53.19 | 56.53 | **59.57** | 58.31 | 56.53 | 58.05 |
| Qwen3-Coder-30A3B | 37.69 | **41.34** | 38.30 | 39.82 | 39.82 | 38.30 | 39.21 |
| Qwen3-4B-Thinking | 63.53 | 60.79 | 62.01 | 62.61 | 62.31 | **64.13** | 63.83 |
| Qwen3-Think-30A3B | 69.30 | 68.39 | 68.69 | 65.65 | 67.78 | 67.48 | **72.04** |

## 5.1 CLOSE-ENDED TASKS: COMPETITIVE WITH MAJORITY VOTING

On AIME and GPQA, THINKMERGE is competitive with majority voting (MV), often matching or slightly exceeding it when the merge is performed across all parallel thoughts ("All-Merge"). For instance, on AIME, Qwen3-4B at $K{=}4$ improves from MV 68.0% to THINKMERGE 72.0% **(+4.0%)**, and Qwen3-14B at $K{=}8$ improves from 74.0 to 78.0 **(+4.0 %)**, shown in Table 1. On GPQA, THINKMERGE at small $K$ is reliably strong: at $K{=}2$, Qwen3-4B improves from 49.2% to 50.3% **(+1.1 %)**, Qwen3-14B from 60.8% to 61.6% **(+0.8 %)**, and R1-Distill-Qwen-7B from 44.9% to 49.2% **(+4.3 %)** (Table 2). When applying Shortest-$K$ Merge strategy, both THINKMERGE and MV are boosted, but MV is generally stronger than THINKMERGE on AIME/GPQA (e.g., AIME Qwen3-14B at $K{=}8$: 80.0% vs. 78.7%; GPQA shows the same trend at $K{=}4,8$). This indicates that when there is a large reasoning pool, for math questions, shortest-$K$ avoiding redundant self-reflection loops is a strong inductive bias to select high-quality solutions, in which THINKMERGE cannot help to much.

**Trimming repeated reflections.** Our regex-based trimming variant (*Ours+Trimming*) shows mixed, model-dependent effects—sometimes helpful (e.g., AIME with R1-Distill-Qwen-7B at $K{=}8$: 48.0%), but often neutral or slightly negative. A sample-by-sample checking indicates that reflection patterns vary widely across model–task combinations, making a single, robust pattern-matching rule difficult to design (and brittle rules risk removing useful content). Consequently, we don't use trimming in the subsequent open-ended experiments.

**Answer-phase temperature.** Lowering the answer-phase temperature $T_{\mathrm{ans}}$ offers *no consistent gain*. On AIME, many cells mildly drop at $T_{\mathrm{ans}}{=}0.3$ especially for $K{=}4,8$ (e.g., All-Merge, Qwen3-4B: $K{=}8$, 70.0%→68.7%), with modest increases on $K{=}2$ (e.g., Shortest Merge, Qwen3-4B:, 65.3%→66.7%) (Table 3). Our takeaway is that: once the thinking phase already induces enough diversity, further *"cooling"* at the answer phase is unnecessary.

## 5.2 OPEN-ENDED CODE: FEWER IS BETTER, AND "SHORTEST" BIAS LOSE EFFECTIVENESS

On LiveCodeBench, THINKMERGE outperforms single-pass baselines, with the most reliable gains at *small* $K$. For DeepCoder-14B-Preview, overall pass@1 improves from 55.32→61.09 **(+5.77 %)** (best at *Shortest-$K$*, $K{=}2$); for Qwen3-8B, it improves from 57.14→59.57 **(+2.43 %)** (best at *All-Merge*, $K{=}2$); see Table 4.

**The *"shortest"* inductive bias is not universal.** Prior math QA reports that "shorter chains are often better" attributing failures to long, looping reflections. In code generation, Shortest-$K$ Merge

Table 5: DeepResearch agent benchmarks Pass@1 (%). Row-wise best is highlighted.

| Benchmark | Model | Baseline | THINKMERGE | | |
|---|---|---|---|---|---|
| | | | N=2 | N=4 | N=8 |
| GAIA | WebSailor-3B | 32.22 | **33.49** | 15.04 | 5.34 |
| | WebSailor-7B | 35.52 | 33.98 | **41.26** | 36.89 |
| | WebSailor-32B | 46.64 | 48.55 | **51.46** | 50.49 |
| Xbench-DeepSearch | WebSailor-3B | 26.40 | **26.80** | 12.20 | 5.40 |
| | WebSailor-7B | 37.80 | 43.20 | **48.00** | 47.20 |
| | WebSailor-32B | 50.40 | 50.20 | 55.20 | **57.60** |
| BrowseComp-EN (200) | WebSailor-3B | 4.70 | **6.30** | 3.50 | 2.50 |
| | WebSailor-7B | 6.30 | 11.00 | **13.60** | 13.10 |
| | WebSailor-32B | 11.80 | 13.10 | 13.40 | **14.50** |
| BrowseComp-ZH | WebSailor-3B | 8.67 | **11.76** | 4.15 | 2.77 |
| | WebSailor-7B | 14.01 | 21.45 | **24.91** | 22.49 |
| | WebSailor-32B | 21.97 | 26.30 | **28.37** | 27.34 |

is *not* always benefit: shorter traces may omit necessary scaffolding (imports, helper functions) and harm executability.

**Helps most on Medium/Hard Questions.** The difficulty split shows that improvements concentrate on *Medium/Hard* (Tables 13–14). On *Hard*, DeepCoder-14B rises 20.69→28.97 **(+8.28 %))** and Qwen3-8B 24.14→31.72 **(+7.58 %))**, while *Easy* is largely saturated (Table 12). Due to space limits, full LiveCodeBench results across difficulty levels (Tables 12–14) are in Appendix A.3.

**How many thoughts to merge?** On *closed-ended* datasets, increasing $K$ generally helps but shows diminishing returns beyond small $K$ and depends on the base model: in several cases $K=4$ already saturates, and $K=8$ doubles the compute but offers little additional gain or may even slightly regress. Consistently, majority voting also shows diminishing returns as $N$ grows on those tasks; Figure 6 in the appendix indicates saturation when $N \geq 8$. For *open-ended* code, the saturation point is even earlier: $K=2$ is typically best, often outperforming $K=4$ and $K=8$. This is good news for practical deployment. The strong performance is achievable with small ensembles, keeping affordable memory and computation costs for online serving.

Finally, we test whether THINKMERGE can also benefit *agentic* deep-research settings, where the model must interleave reasoning (e.g., `<think>...</think>`) and tool calls before producing an answer. Concretely, we evaluate three Tongyi-WebSailor agents—WebSailor-3B, WebSailor-7B, and WebSailor-32B—on four challenging web-based benchmarks: **BrowseComp-en** (Wei et al., 2025), **BrowseComp-zh** (Zhou et al., 2025), **GAIA** (Mialon et al., 2023), and **XbenchDeepSearch** (Chen et al., 2025a). BrowseComp-en/zh focus on hard-to-find, multi-hop factual queries in English and Chinese. Because BrowseComp-en is large (1,266 questions in total), we randomly sample 200 questions as a test subset, making it comparable in size to BrowseComp-zh (289 questions). GAIA requires robust tool use for multi-step real-world tasks; following prior work (Li et al., 2025b), we evaluate on the 103 text-only validation cases. XbenchDeepSearch targets professional-style, deep information retrieval. We use the WebSailor agent pipeline with the recommended decoding hyperparameters: temperature 0.6, top-p 0.95, and context length 32,768.

## 5.3 OPEN-ENDED DEEPRESEARCH AGENTS

Finally, we test whether THINKMERGE can also benefit *agentic* deep-research settings, where the model must interleave internal reasoning (e.g., `<think>...</think>`) with tool calls before producing an answer. We evaluate three Tongyi-WebSailor agents—WebSailor-3B, WebSailor-7B, and WebSailor-32B—on four challenging web-based benchmarks: **BrowseComp-en** (Wei et al., 2025), **BrowseComp-zh** (Zhou et al., 2025), **GAIA** (Mialon et al., 2023), and **XbenchDeepSearch** (Chen et al., 2025a). BrowseComp-en/zh focus on hard-to-find, multi-hop factual queries in English and Chinese. Because BrowseComp-en is large (1,266 questions in total), we randomly sample 200 questions as a test subset, making it comparable in size to BrowseComp-zh (289 questions). GAIA requires robust tool use on multi-step real-world tasks; following prior work (Li et al., 2025b), we evaluate on the 103 text-only validation cases. XbenchDeepSearch targets professional-style, deep information retrieval. We use the WebSailor agent pipeline (Alibaba-NLP, 2025) with the recommended decoding hyperparameters: temperature 0.6, top-p 0.95, and context length 32,768. For computational efficiency, we replace the evaluator model Qwen2.5-72B with GPT-4.1 and swap the Google Search API for the Serper API (Serper.dev) to reduce API fee costs; under this configuration, we re-run all baselines and report the average over five runs in Table 5.

Table 6: Performance of Majority Voting, THINKMERGE, and Prob-Merge on AIME'25 across different numbers of samples $K$.

| Model | Majority Voting | THINKMERGE | Prob-Merge |
|---|---|---|---|
| $K = 2$ | | | |
| Qwen3-4B | 63.3 | **66.7** | 62.0 |
| Qwen3-14B | 70.7 | **72.0** | 68.0 |
| R1-Distill-Qwen-7B | 40.7 | **41.3** | 34.7 |
| $K = 4$ | | | |
| Qwen3-4B | 68.0 | **72.0** | 65.4 |
| Qwen3-14B | **73.3** | 72.0 | 70.0 |
| R1-Distill-Qwen-7B | **47.3** | 46.0 | 32.7 |
| $K = 8$ | | | |
| Qwen3-4B | 68.7 | **70.0** | 62.0 |
| Qwen3-14B | 74.0 | **78.0** | 69.4 |
| R1-Distill-Qwen-7B | **46.7** | 45.3 | 34.0 |

**Scaling up the agent makes test-time ensembles effective.** For the stronger 7B and 32B Web-Sailor agents, THINKMERGE consistently improves over the single-run baseline, often by a large margin. On XbenchDeepSearch, WebSailor-32B improves from $50.4$ to $57.6$ at $N{=}8$ **(+7.2)**, while WebSailor-7B rises from $37.8$ to $48.0$ **(+10.2)**. On GAIA, WebSailor-32B reaches $51.46$ at $N{=}4$, and WebSailor-7B improves from $35.52$ to $41.26$. The two BrowseComp benchmarks show a similar pattern. These results indicate that, once the underlying agent is sufficiently capable, running multiple research trajectories in parallel and then merging their answers is an effective way to trade test-time compute for higher performance.

In contrast, the 3B WebSailor agent only benefits from THINKMERGE at *small* $N$: across all four benchmarks, $N{=}2$ yields mild gains, but performance degrades noticeably at $N{=}4, 8$. This is consistent with a "garbage in, garbage out" intuition: when most trajectories are low-quality or off-topic, ensembling more of them will not fix the errors and can even dilute the few good traces. Qualitatively, small models tend to generate many such weak research trajectories, so averaging over too many of them "washes out" the good ones, suggesting that aggressive test-time compute scaling is only beneficial beyond a certain capability threshold.

## 5.4 ABLATION: MERGE LOGIT VS. MERGE PROBABILITY FOR REASONING MODELS

Our THINKMERGE aggregates decoding at the *logit* level: at each answer-time decoding step $t$, we take the arithmetic mean over the pre-softmax logits as described in Section 4. A natural alternative, more in line with prior work (Wicks et al., 2025; Xu et al., 2024; Guo et al., 2024) on token probability ensembling, is to first normalize each logit vector and then average probabilities: $p_t = \frac{1}{K} \sum_k \mathrm{softmax}(z_t^{(k)})$. We refer to this variant as *Prob-Merge*. In both cases, aggregation is restricted to the answer phase; the `<think>` phase remains fully independent.

Table 6 compares Majority Voting (MV), THINKMERGE, and Prob-Merge on AIME'25 for three reasoning models and different numbers of samples $K$. Across all configurations, Prob-Merge is consistently weaker than THINKMERGE (logit-level merging) and often even underperforms MV, especially for the weaker R1-Distill-Qwen-7B model, where performance degrades sharply as $K$ grows (e.g., $47.3$ for MV vs. $32.7$ for Prob-Merge at $K{=}4$). In contrast, logit-level DirectMerge either matches or improves upon majority voting in most settings. These trends suggest that, for reasoning models, aggregating *before* normalization is more robust than averaging already-normalized probabilities. Besides, from an implementation perspective, merging over probabilities also conflicts with the standard logit-processor interface (e.g., top-$k$/top-$p$ filtering) in modern inference frameworks such as vLLM, which operate directly on logits.

## 6 CONCLUSION

In this work, we introduce THINKMERGE, a training-free parallel test-time scaling for open-ended reasoning. Given a prompt, we sample $K$ diverse reasoning traces up to a delimiter, then decode a single answer by averaging next-token logits across traces at every step; the chosen token is fed back to all contexts so the ensemble continues to guide subsequent tokens. THINKMERGE preserves compatibility with standard decoding controls and integrates naturally with modern inference stacks (e.g., vLLM, SGLang), making it easy to deploy for both online serving and offline batch decoding. Empirically, on closed-ended math/science QA, the proposed method is competitive with, and sometimes exceeds, majority voting. On open-ended tasks, LiveCodeBench and deep research tasks THINKMERGE improves overall pass@1 of several models without additional training.

ACKNOWLEDGMENTS

This material is based upon work supported by the Air Force Office of Scientific Research under award number FA2386-24-1-4011, and this research is partially supported by the Singapore Ministry of Education Academic Research Fund Tier 1 (Award No. T1 251RES2509).

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

# A   MORE EXPRIMENT DETAILS

## A.1   SAMPLING HYPER-PARAMETERS

For *closed-ended* benchmarks across all tested models, we set temperature=0.6 and top-p=1.0 (i.e., no probabilistic truncation). For *open-ended* LiveCodeBench, we use model-specific settings: for Qwen/Qwen3-Coder-30B-A3B-Instruct , we set temperature=0.7 and top-p=0.8 (because it is a non-reasoning model). For all other models, we set temperature=0.6 and top-p=0.95.

## A.2   EXPERIMENTAL RESULTS WITH STANDARD DEVIATION

Table 7: Baseline (Majority Voting) on AIME 2025. Results are mean $\pm$ std. For each $n$, majority@$K$ is evaluated under three combination strategies.

| Model | Direct-Voting | Early-Ready | Shortest-$K$ Merge |
|---|---|---|---|
| $K = 2$ | | | |
| Qwen3-4B | $0.633 \pm 0.047$ | $0.667 \pm 0.047$ | $0.707 \pm 0.032$ |
| Qwen3-14B | $0.707 \pm 0.025$ | $0.720 \pm 0.034$ | $0.753 \pm 0.045$ |
| DeepSeek-R1-Distill-Qwen-7B | $0.407 \pm 0.049$ | $0.407 \pm 0.049$ | $0.507 \pm 0.025$ |
| $K = 4$ | | | |
| Qwen3-4B | $0.680 \pm 0.034$ | $0.727 \pm 0.033$ | $0.753 \pm 0.017$ |
| Qwen3-14B | $0.733 \pm 0.021$ | $0.760 \pm 0.025$ | $0.780 \pm 0.034$ |
| DeepSeek-R1-Distill-Qwen-7B | $0.473 \pm 0.044$ | $0.473 \pm 0.044$ | $0.520 \pm 0.016$ |
| $K = 8$ | | | |
| Qwen3-4B | $0.687 \pm 0.054$ | $0.733 \pm 0.047$ | $0.753 \pm 0.017$ |
| Qwen3-14B | $0.740 \pm 0.025$ | $0.774 \pm 0.013$ | $0.800 \pm 0.021$ |
| DeepSeek-R1-Distill-Qwen-7B | $0.467 \pm 0.021$ | $0.467 \pm 0.021$ | $0.527 \pm 0.033$ |

Table 8: THINKMERGE on AIME 2025. Results are mean $\pm$ std under four combination strategies.

| Model | Direct-Merge | Suffix Trimming | Early-Ready | Shortest-$K$ Merge |
|---|---|---|---|---|
| $K = 2$ | | | | |
| Qwen3-4B | $0.667 \pm 0.059$ | $0.660 \pm 0.068$ | $0.667 \pm 0.027$ | $0.653 \pm 0.017$ |
| Qwen3-14B | $0.720 \pm 0.016$ | $0.727 \pm 0.033$ | $0.733 \pm 0.021$ | $0.747 \pm 0.034$ |
| DeepSeek-R1-Distill-Qwen-7B | $0.413 \pm 0.034$ | $0.420 \pm 0.034$ | $0.413 \pm 0.034$ | $0.480 \pm 0.050$ |
| $K = 4$ | | | | |
| Qwen3-4B | $0.720 \pm 0.016$ | $0.680 \pm 0.034$ | $0.727 \pm 0.033$ | $0.693 \pm 0.025$ |
| Qwen3-14B | $0.720 \pm 0.034$ | $0.733 \pm 0.021$ | $0.733 \pm 0.021$ | $0.760 \pm 0.025$ |
| DeepSeek-R1-Distill-Qwen-7B | $0.460 \pm 0.033$ | $0.460 \pm 0.039$ | $0.453 \pm 0.027$ | $0.507 \pm 0.033$ |
| $K = 8$ | | | | |
| Qwen3-4B | $0.700 \pm 0.052$ | $0.687 \pm 0.016$ | $0.708 \pm 0.043$ | $0.727 \pm 0.033$ |
| Qwen3-14B | $0.780 \pm 0.045$ | $0.733 \pm 0.021$ | $0.780 \pm 0.034$ | $0.787 \pm 0.016$ |
| DeepSeek-R1-Distill-Qwen-7B | $0.453 \pm 0.045$ | $0.480 \pm 0.054$ | $0.453 \pm 0.034$ | $0.527 \pm 0.025$ |

Table 9: Temperature study of THINKMERGE on AIME: parallel thinking uses the officially suggested temperature; the answer phase uses a smaller $T{=}0.3$. Results are mean $\pm$ std.

| Model | Direct-Merge | Early-Ready | Shortest-$K$ Merge |
|---|---|---|---|
| $K = 2$ | | | |
| Qwen3-4B | $0.647 \pm 0.062$ | $0.673 \pm 0.039$ | $0.667 \pm 0.021$ |
| Qwen3-14B | $0.713 \pm 0.026$ | $0.733 \pm 0.021$ | $0.740 \pm 0.033$ |
| DeepSeek-R1-Distill-Qwen-7B | $0.413 \pm 0.027$ | $0.413 \pm 0.027$ | $0.487 \pm 0.054$ |
| $K = 4$ | | | |
| Qwen3-4B | $0.700 \pm 0.037$ | $0.720 \pm 0.034$ | $0.707 \pm 0.032$ |
| Qwen3-14B | $0.727 \pm 0.025$ | $0.727 \pm 0.033$ | $0.753 \pm 0.034$ |
| DeepSeek-R1-Distill-Qwen-7B | $0.427 \pm 0.039$ | $0.433 \pm 0.042$ | $0.520 \pm 0.016$ |
| $K = 8$ | | | |
| Qwen3-4B | $0.687 \pm 0.034$ | $0.693 \pm 0.039$ | $0.707 \pm 0.025$ |
| Qwen3-14B | $0.747 \pm 0.017$ | $0.767 \pm 0.021$ | $0.773 \pm 0.025$ |
| DeepSeek-R1-Distill-Qwen-7B | $0.447 \pm 0.034$ | $0.440 \pm 0.025$ | $0.513 \pm 0.034$ |

Table 10: GPQA — Baseline (Majority Voting). Results are mean $\pm$ std. For each $n$, majority@$n$ is evaluated under three combination strategies.

| Model | Direct-Voting | Early-Ready | Shortest-K Merge |
|---|---|---|---|
| $K = 2$ | | | |
| Qwen3-4B | $0.492 \pm 0.020$ | $0.494 \pm 0.020$ | $0.520 \pm 0.013$ |
| Qwen3-14B | $0.608 \pm 0.015$ | $0.608 \pm 0.015$ | $0.634 \pm 0.021$ |
| DeepSeek-R1-Distill-Qwen-7B | $0.449 \pm 0.025$ | $0.474 \pm 0.009$ | $0.449 \pm 0.011$ |
| $K = 4$ | | | |
| Qwen3-4B | $0.512 \pm 0.022$ | $0.514 \pm 0.018$ | $0.537 \pm 0.011$ |
| Qwen3-14B | $0.640 \pm 0.012$ | $0.640 \pm 0.012$ | $0.652 \pm 0.007$ |
| DeepSeek-R1-Distill-Qwen-7B | $0.503 \pm 0.015$ | $0.503 \pm 0.013$ | $0.502 \pm 0.010$ |
| $K = 8$ | | | |
| Qwen3-4B | $0.533 \pm 0.025$ | $0.533 \pm 0.025$ | $0.555 \pm 0.013$ |
| Qwen3-14B | $0.639 \pm 0.011$ | $0.641 \pm 0.011$ | $0.659 \pm 0.008$ |
| DeepSeek-R1-Distill-Qwen-7B | $0.525 \pm 0.021$ | $0.525 \pm 0.020$ | $0.529 \pm 0.019$ |

Table 11: GPQA — Our method THINKMERGE. Results are mean $\pm$ std under four combination strategies.

| Model | Direct-Merge | Suffix Trimming | Early-Ready | Shortest-$K$ Merge |
|---|---|---|---|---|
| $K = 2$ | | | | |
| Qwen3-4B | $0.503 \pm 0.016$ | $0.500 \pm 0.018$ | $0.503 \pm 0.016$ | $0.526 \pm 0.006$ |
| Qwen3-14B | $0.616 \pm 0.008$ | $0.614 \pm 0.008$ | $0.621 \pm 0.004$ | $0.637 \pm 0.010$ |
| DeepSeek-R1-Distill-Qwen-7B | $0.492 \pm 0.020$ | $0.492 \pm 0.020$ | $0.490 \pm 0.022$ | $0.438 \pm 0.018$ |
| $K = 4$ | | | | |
| Qwen3-4B | $0.522 \pm 0.008$ | $0.522 \pm 0.009$ | $0.524 \pm 0.007$ | $0.518 \pm 0.009$ |
| Qwen3-14B | $0.630 \pm 0.012$ | $0.627 \pm 0.014$ | $0.624 \pm 0.010$ | $0.638 \pm 0.013$ |
| DeepSeek-R1-Distill-Qwen-7B | $0.489 \pm 0.010$ | $0.490 \pm 0.009$ | $0.489 \pm 0.010$ | $0.474 \pm 0.016$ |
| $K = 8$ | | | | |
| Qwen3-4B | $0.516 \pm 0.017$ | $0.513 \pm 0.015$ | $0.513 \pm 0.022$ | $0.538 \pm 0.013$ |
| Qwen3-14B | $0.641 \pm 0.013$ | $0.630 \pm 0.011$ | $0.641 \pm 0.011$ | $0.637 \pm 0.011$ |
| DeepSeek-R1-Distill-Qwen-7B | $0.500 \pm 0.021$ | $0.500 \pm 0.021$ | $0.502 \pm 0.022$ | $0.472 \pm 0.009$ |

### A.3 LiveCodeBench Easy / Medium / Hard Level Performance

Table 12: LiveCodeBench *Easy* Pass@1 (%). Row-wise best among merge settings is highlighted. The best score within each category is **bold**; the tier is underlined.

| Model | Baseline | Direct-Merge | | | Shortest-$K$ Merge | | |
|---|---|---|---|---|---|---|---|
| | | $K=8$ | $K=4$ | $K=2$ | $K=8$ | $K=4$ | $K=2$ |
| DeepCoder-14B-Preview | 98.77 | **98.77** | 96.30 | 97.53 | 97.53 | **98.77** | 97.53 |
| Qwen3-8B | 97.53 | 95.06 | **98.77** | 97.53 | 96.30 | 95.06 | **97.53** |
| Qwen3-Coder-30A3B | 90.12 | **93.83** | 90.12 | 87.65 | 90.12 | 88.89 | **93.83** |
| Qwen3-4B-Thinking | 98.77 | 98.77 | 98.77 | 97.53 | 97.53 | 97.53 | 97.53 |
| Qwen3-Think-30A3B | 98.77 | 98.77 | 98.77 | 97.53 | 98.77 | 98.77 | 98.77 |

Table 13: LiveCodeBench *Medium* Pass@1. Row-wise best among merge settings is highlighted.

| Model | Baseline | Direct-Merge | | | Shortest-$K$ Merge | | |
|---|---|---|---|---|---|---|---|
| | | $K=8$ | $K=4$ | $K=2$ | $K=8$ | $K=4$ | $K=2$ |
| DeepCoder-14B-Preview | 69.90 | 66.02 | 71.84 | 75.73 | 76.70 | 75.73 | **77.67** |
| Qwen3-8B | 71.84 | 63.11 | 66.99 | 68.93 | **69.75** | 67.96 | 66.99 |
| Qwen3-Coder-30A3B | 36.89 | **42.72** | 34.95 | 41.75 | 41.75 | 38.83 | 37.86 |
| Qwen3-4B-Thinking | 76.70 | 69.90 | 75.73 | **76.70** | 72.82 | **76.70** | 75.73 |
| Qwen3-Think-30A3B | 82.52 | 81.55 | **84.47** | 80.58 | 78.64 | 77.67 | **84.47** |

Table 14: LiveCodeBench *Hard* Pass@1 (%). Row-wise best among merge settings is highlighted.

| Model | Baseline | Direct-Merge | | | Shortest-$K$ Merge | | |
|---|---|---|---|---|---|---|---|
| | | $K=8$ | $K=4$ | $K=2$ | $K=8$ | $K=4$ | $K=2$ |
| DeepCoder-14B-Preview | 20.69 | 25.52 | 24.83 | 24.14 | 26.21 | 26.90 | **28.97** |
| Qwen3-8B | 24.14 | 22.76 | 25.52 | **31.72** | 28.97 | 26.90 | 29.66 |
| Qwen3-Coder-30A3B | 8.97 | 11.03 | **11.72** | **11.72** | 10.34 | 9.66 | 9.66 |
| Qwen3-4B-Thinking | 34.48 | 33.10 | 31.72 | 33.10 | 35.17 | **36.55** | **36.55** |
| Qwen3-Think-30A3B | 43.45 | 42.07 | 40.69 | 37.24 | 42.76 | 42.76 | **48.28** |

## A.4 CLOSE-ENDED TASKS: MAJORITY VOTING SATURATES QUICKLY WITH N

Figure 6: On Close-edned task AIME'25 and GPQA, majority voting saturates quickly with $N$.

## B LLM USAGE

We used large language models (ChatGPT and Gemini) as writing and formatting assistants. In particular, it helped refine grammar and phrasing, improve clarity, and suggest edits to figure/table captions and layout (e.g., column alignment, caption length, placement). The LLM did not contribute to research ideation, experimental design, implementation, data analysis, or technical content beyond surface-level edits. All outputs were reviewed and edited by the authors, who take full responsibility for the final text and visuals.

