# OpenReview forum: "Think in Parallel, Answer as One: Logit Averaging for Open-Ended Reasoning"
_ICLR.cc/2026/Conference — ICLR 2026 Poster_

### Official Review · Reviewer_SGtm · 2025-10-14

**Soundness:** 2
**Presentation:** 2
**Contribution:** 1
**Rating:** 2
**Confidence:** 5

**Summary:**

This paper proposes a new way to deal with open-ended questions with LLM, named ThinkMerge, in which authors used majority voting, map-reduce, etc.

**Strengths:**

The paper is well structured and clearly written. I was able to easily understand the ideas the authors intended to convey.

**Weaknesses:**

I believe the primary issue lies in the lack of novelty. Although the paper claims to focus on open-ended questions, the datasets used, such as AIME and GPQA, are predominantly composed of closed-ended questions. This mismatch undermines the validity of the problem statement from the outset.

Secondly, the proposed method, described as "think merge," appears to be essentially a form of majority voting. The use of a map-and-reduce framework is also not new, as similar approaches can be found in many prior works, such as GraphRAG.

In terms of experiments, I don’t see any significant improvement on genuinely open-ended questions. Overall, in its current form, I don’t find the paper acceptable.

**Questions:**

I have no questions, as the writing is very clear.

---

> ### Author Response · Authors · 2025-11-21
> **Rebuttal by Authors [1/2]**
>
> Thank you for your valuable review and suggestions. Below we respond to the comments in **Weaknesses (W)**.
>
> ---
>
> ***W1: I believe the primary issue lies in the lack of novelty. Although the paper claims to focus on open-ended questions, the datasets used, such as AIME and GPQA, are predominantly composed of closed-ended questions.***
>
> This comment touches on two aspects: (i) lack of novelty, and (ii) limited evaluation on open-ended questions. Since the "lack of novelty" point overlaps substantially with W2, here we focus on the open-ended evaluation, and we address novelty explicitly in our response to W2.
>
> Our intended scope is to study open-ended reasoning under the modern "think–then–answer" paradigm. The open-ended setting is where we believe ThinkMerge is most useful, since solution-level voting is ill-defined. Already in the main paper, we evaluate on LiveCodeBench, where correctness is measured by program execution and pass@1 cannot be improved by majority voting over free-form outputs. On the hard subset of LiveCodeBench, ThinkMerge improves pass@1 by +8.28 for DeepCoder-14B-Preview and +7.58 for Qwen3-8B, as reported in the abstract and main text.
>
>
> In the revised version we further strengthen the open-ended evaluation by **adding four agent deep-research benchmarks** (BrowseComp-en, BrowseComp-zh, GAIA, XBenchDeepSearch) with three Tongyi-WebSailor agent models, where the model must interleave long-form reasoning with tool calls and answers are graded by task-specific success metrics. On these genuinely open-ended tasks, ThinkMerge consistently improves over the single-run baseline across 7B and 32B agents (see the new $\\textrm{\\color{blue}Section 5.4}$). For example, WebSailor-32B improves from $50.4$ to $57.6$ at $N{=}8$ **(+7.2)**, while WebSailor-7B rises from $37.8$ to $48.0$ **(+10.2)**. On GAIA, WebSailor-32B reaches $51.46$ at $N{=}4$ **(+4.82)**, and WebSailor-7B improves from $35.52$ to $41.26$ **(+5.74)**. The two BrowseComp benchmarks exhibit a similar trend.
>
>
> ---
>
> ***W2: Secondly, the proposed method, described as "think merge", appears to be essentially a form of majority voting. The use of a map-and-reduce framework is also not new, as similar approaches can be found in many prior works, such as GraphRAG.***
>
>
> We **respectfully but disagree** with the claim that ThinkMerge is "essentially a form of majority voting". This characterization comes from an overly coarse view of aggregation: under such a view, almost any method that combines multiple candidates could be called "majority voting" – including many works the reviewer CwCp suggested [1,2,3,4,5].
>
> Regarding the "map-and-reduce" comment, we agree that this is a generic high-level pattern. GraphRAG, for instance, uses map–reduce on the retrieval then summarization side: it builds a graph over documents, produces community-level summaries and partial answers in parallel, and then calls an LLM once more to summarize these partial answers into a final response. We refine the related work section to include the discussion of GraphRAG [6].
>
> Regarding novelty:
>
> Our work doesn't seek "unnecessary" novelty. Instead, we contribute to a simple and scalable thinking merging framework that is applicable to a wide range of open-ended scenarios and includes off-the-shelf code implementation. Our method uses a simple recipe to achieve superior performance compared to previous work, and this is the "empirical novelty" that we believe is truly meaningful to the community.

---

> > ### Author Response · Authors · 2025-11-21
> > **Rebuttal by Authors [2/2]**
> >
> > ---
> >
> > ***W3: In terms of experiments, I don’t see any significant improvement on genuinely open-ended questions. Overall, in its current form, I don’t find the paper acceptable.***
> >
> > In the original submission, our open-ended evaluation focused on LiveCodeBench, where correctness is judged by *program execution* and majority voting over free-form outputs is not applicable. On the hard subset of LiveCodeBench, ThinkMerge already improves pass@1 by **+7.58** for Qwen3-8B and **+8.28** for DeepCoder-14B-Preview. We agree, however, that these results alone may not fully convince the reader that the method is broadly useful for open-domain reasoning.
> >
> >
> > To address this, the revised version substantially **expands the open-ended evaluation** by adding a new subsection on **DeepResearch agents** and a new $\\textrm{\\color{blue}Table 5}$. We integrate ThinkMerge into the Tongyi-WebSailor web-browsing agent and evaluate on four challenging **open-domain deep research** benchmarks: GAIA, Xbench-DeepSearch, BrowseComp-EN, and BrowseComp-ZH. These tasks require multi-step tool use and long-form answers (often graded by success@1), making them representative of the "genuinely open-ended" scenarios highlighted in the review.
> >
> > The new results show **clear and consistent gains** for sufficiently strong agents:
> >
> > > * On **GAIA**, WebSailor-7B improves from **35.52** (baseline) to **41.26** at $N{=}4$, and WebSailor-32B improves from **46.64** to **51.46**.
> > > * On **Xbench-DeepSearch**, WebSailor-7B goes from **37.80** to **48.00** at $N{=}4$, and WebSailor-32B from **50.40** to **57.60** at $N{=}8$.
> > > * On **BrowseComp-EN**, WebSailor-7B improves from **6.30** to **13.60** at $N{=}4$, and WebSailor-32B from **11.80** to **14.50**.
> > > * On **BrowseComp-ZH**, WebSailor-7B rises from **14.01** to **24.91** at $N{=}4$, and WebSailor-32B from **21.97** to **28.37**.
> > >  These are **substantial absolute gains** (often 5–10 points) on demanding, real-world style open-ended tasks.
> >
> > At the same time, the new table also reveals an important **limitation**: for the weaker WebSailor-3B agent, ThinkMerge only helps at very small $N$ (e.g., GAIA and BrowseComp yield modest gains at $N{=}2$), while larger $N$ can hurt performance as highly noisy traces are merged. We explicitly discuss this behavior in the revision: ThinkMerge is most effective when the underlying agent is reasonably capable, and we do *not* claim universal gains for arbitrarily weak models or very large $K$.
> >
> > Taken together, the expanded LiveCodeBench and DeepResearch experiments show that, contrary to the initial impression of "no significant improvement", ThinkMerge can deliver **meaningful and practically relevant gains** on genuinely open-ended reasoning and deep-research tasks, especially for modern, moderately strong agents (7B–32B). We will make this positioning clearer in the main text to better communicate where the method is most beneficial and where its limitations lie.
> >
> > ---
> >
> > ***References:***
> >
> > [1]: Product of Experts - Geoff Hinton
> >
> > [2]: M-Ped: Multi-Prompt Ensemble Decoding for Large Language Models - Guo et.al, CORR 2024
> >
> > [3]: Bridging the Gap between Different Vocabularies for LLM Ensemble - NAACL 2024
> >
> > [4]: Token-level Ensembling of Models with Different Vocabularies - Rachel Wicks et.al (arxiv)
> >
> > [5]: Llm-blender: Ensembling large language models with pairwise ranking and generative fusion - Jiang et.al
> >
> > [6] Edge, Darren, et al. "From local to global: A graph rag approach to query-focused summarization." arXiv preprint arXiv:2404.16130 (2024).

---

### Official Review · Reviewer_bs9g · 2025-10-31

**Soundness:** 3
**Presentation:** 3
**Contribution:** 3
**Rating:** 6
**Confidence:** 4

**Summary:**

The paper proposes THINKMERGE, a training-free way to make parallel chain-of-thought work for open-ended tasks where majority voting can’t, by averaging token-level logits from K parallel traces after the reasoning delimiter and decoding a single shared answer. On AIME/GPQA it matches or slightly improves over standard parallel CoT with voting, and on LiveCodeBench it gives clear pass@1 gains where voting fails. I’m slightly reserved about the reliance on logit access / synchronized delimiters and the fact that some gains on closed-ended tasks are modest, but the core idea is clean and the evaluation is broad enough to justify it.

I would have preferred if the code was supplied to have a hand-on experiment with the approach, but I have not had time to reimplement it, however, theoritically it is intact.

**Strengths:**

- Addresses a real gap: parallel CoT doesn’t work for open-ended outputs; this makes it usable there.
- token-level logit averaging after a shared delimiter, easy to plug into vLLM/SGLang.
- Evaluates on both closed-ended (AIME, GPQA) and open-ended (LiveCodeBench), so it’s not overfitted to math-only.
- Keeps contexts synchronized while decoding, which is a nice engineering detail for practicality.

**Weaknesses:**

- This looks to be delimiter-dependent. It assumes clean think/answer separation; models that ramble or reflect will hurt alignment.
- On AIME/GPQA it sometimes only matches or even loses to majority voting.
- I’m skeptical about pure logit averaging, since it can dilute a minority-but-correct trace. A simple reweighting or confidence-based scheme might help; at minimum I’d like to see an ablation showing how often correct traces get downvoted by the ensemble
- The method gives clear gains at small K (e.g. K=2), but increasing K further doesn’t reliably improve pass@1 and can even regress, likely because divergent traces interfere when logits are averaged. I think the approach is effective, but scaling k for more benefit is not working as expected.

**Questions:**

Please refer to weaknesses

---

> ### Author Response · Authors · 2025-11-21
> **Rebuttal by Authors**
>
> Thank you for your supportive review and suggestions. Below we respond to the comments in **Weaknesses (W)**.
>
> ---
>
> ***W1: This looks to be delimiter-dependent. It assumes clean think/answer separation; models that ramble or reflect will hurt alignment.***
>
> We agree that the current formulation assumes an explicit separation between the “thinking” and “answer” phases. This matches the emerging practice in modern reasoning LMs: after DeepSeek-R1 [1], many widely used models (e.g., the Qwen3 series [2]) adopt a similar ```<think>…</think>```–style paradigm that cleanly delimits internal reasoning from the final answer. In such models, “rambling” or extended reflection typically happens inside the ```<think>``` block rather than after the answer.
>
>
> In our work, we explicitly study robustness to noisy thinking segments via the "(C) Trimming (De-Repeat Suffix)" heuristic in $\\textrm{\\color{blue}Section 4.2}$, which attempts to clean up repeated or low-information tails in the thinking part before aggregation. As shown in $\\textrm{\\color{blue}Tables 1}$ and $\\textrm{\\color{blue}2}$, this trimming provides only modest additional gains, suggesting that ThinkMerge is already reasonably robust to imperfect think/answer boundaries in practice. We will clarify in the revision that (i) the method is targeted at the widely adopted think–then–answer paradigm, and (ii) it does not rely on perfectly clean delimiters to work well.
>
> ---
>
> ***W2: On AIME/GPQA it sometimes only matches or even loses to majority voting.***
>
> Our primary target is **open-ended** reasoning, where majority voting over complete solutions is ill-defined or brittle. On closed-ended benchmarks such as AIME and GPQA, majority voting is a well-established baseline, so it is not surprising that ThinkMerge sometimes matches and sometimes slightly lags behind voting.
>
> To better demonstrate the strengths of ThinkMerge in the setting it is designed for, the revised version adds experiments on four **agentic, open-ended** web deep-research benchmarks (BrowseComp-en, BrowseComp-zh, GAIA, and XBenchDeepSearch) ($\\textrm{\\color{blue}Section 5.4}$), where ThinkMerge yields consistent improvements.
>
> ---
>
> ***W3: I’m skeptical about pure logit averaging, since it can dilute a minority-but-correct trace. A simple reweighting or confidence-based scheme might help; at minimum I’d like to see an ablation showing how often correct traces get downvoted by the ensemble.***
>
> We think this is a good point, incorporating confidence-based weights on traces could further improve ThinkMerge. But designing a reliable, task-agnostic confidence score and turning it into stable weights is non-trivial and beyond the scope of this work. In this paper we therefore focus on the simplest uniform logit averaging and leave confidence-weighted variants as future work.
>
>
> ---
>
>
> ***W4: The method gives clear gains at small K (e.g. K=2), but increasing K further doesn’t reliably improve pass@1 and can even regress, likely because divergent traces interfere when logits are averaged. I think the approach is effective, but scaling k for more benefit is not working as expected.***
>
>
> The gains tend to saturate beyond moderate $K$. Empirically, we find that most of the improvement is already captured with small ensembles (e.g., $K{=}4$), after which additional trajectories are often either redundant or highly divergent, which can interfere when their logits are merged. From a practical standpoint, we do not view this saturation as a drawback: it means that practitioners can obtain most of the benefit of ThinkMerge with only a small multiplicative increase in test-time compute, making ThinkMerge amenable to online serving.
>
>
> ---
>
> ***References:***
>
> [1] Guo, Daya, et al. "Deepseek-r1: Incentivizing reasoning capability in llms via reinforcement learning." arXiv preprint arXiv:2501.12948 (2025).
>
> [2] Yang, An, et al. "Qwen3 technical report." arXiv preprint arXiv:2505.09388 (2025).

---

### Official Review · Reviewer_CwCp · 2025-11-01

**Soundness:** 2
**Presentation:** 3
**Contribution:** 2
**Rating:** 4
**Confidence:** 4

**Summary:**

This paper introduces ThinkMerge, an alternate to majority voting paradigm. Instead of doing full rollouts and then majority vote, the authors propose doing parallel rollouts only for the "thinking" tokens, and then decoding from there on by averaging the token logits at each auto-regressive step, and sampling the next token from the resulting distribution. The proposed method neatly fits into widely used inference engines such as vLLM and SGLang, and is compatible with other decoding choices such as top-p/top-k/penalty etc.

**Strengths:**

Strengths:
- The proposed method is quite simple to implement, doesn't require any training, and practically deployable on popular inference engines.
- The technique addresses a limitation of majority voting in open domains, where the response is free form text (code/reports etc) where "majority vote" is ill-defined.
- The four ablations seem well-motivated, and grounded in practical considerations.

**Weaknesses:**

Weaknesses:
- My major concern is with the limited innovation/novelty. The proposed technique is a straight forward application of product of experts ensembling (See [1]). Within the context of application in LLMs, there is prior work in this area:
[2] applies the exact same technique of token fusion in the context of having several prompt variants. They too generate logits by processing each prompt and then averaging the logits at each auto-regressive step. ThinkMerge can be thought of as a special case of this where each prompt is a reasoning trace generated before token fusion. A more general problem of fusion across LLMs with different vocabularies has also been explored by [3, 4]. So the novelty seems to be limited to applying this technique with parallel `<think>...</think>` generations.

- The proposed fusion is naive, and can potentially fail in scenarios where the reasoning traces diverge too much. Suppose, that across parallel generations, the model explores entirely different strategies to solving the problem, the ensembling can collapse. This is not sufficiently explored in the paper.

- Token probability ensembling literature needs to be covered more in Related section [2,3,4,5].

- While the paper emphasizes open domain tasks, it's experimentally validated only with a single benchmark, LiveCodeBench.


[1]: Product of Experts - Geoff Hinton
[2]: M-Ped: Multi-Prompt Ensemble Decoding for Large Language Models - Guo et.al, CORR 2024
[3]: Bridging the Gap between Different Vocabularies for LLM Ensemble - NAACL 2024
[4]: Token-level Ensembling of Models with Different Vocabularies - Rachel Wicks et.al (arxiv)
[5]: Llm-blender: Ensembling large language models with pairwise ranking and generative fusion - Jiang et.al

**Questions:**

Suggestions:

- This work can be a good empirical contribution but would require some more work, especially around analysing which scenarios benefit from fusion approach and investigating it's limitations and failure modes. This is important for the ICLR communities oriented towards practicable solutions.
- Since the emphasis is on open-domain reasoning, include more benchmarks such as Terminal Bench, other coding benchmarks etc.

---

> ### Author Response · Authors · 2025-11-21
> **Rebuttal by Authors [1/2]**
>
> Thank you for your constructive review and suggestions. Below we respond to the comments in **Weaknesses (W)** and **Questions (Q)**.
>
> ---
>
> ***W1: My major concern is with the limited innovation/novelty.***
>
> Thank you for highlighting the relevant literature on ensemble decoding. We have added these works to the revised Related Work section ($\\textrm{\\color{blue}Section 2}$). Here we clarify the distinction between prior methods and our contribution.
>
> First, M-Ped [2] operates at the **probability level**: it averages post-softmax probabilities at each decoding step (Eqs. (2–3) in [2]). The same is true for [2], [3], and [4], all of which ensemble probabilities, not logits. In contrast, our ThinkMerge performs fusion directly at the **logit level**. This difference may appear minor, but it has important empirical and systems-level implications, as described below:
>
> - Empirically, $\\textrm{\\color{blue}Section 5.5}$ of our paper revision compares "merge logits" and "merge probabilities" on AIME 25. We find that probability-level merging performs *consistently worse* than simple majority voting, whereas logit-level merging provides substantial gains. This demonstrates that directly reusing probability-level fusion from prior work is insufficient for complex reasoning, and that our logit-based variant is not a cosmetic modification.
> - From a systems perspective, probability-level merging conflicts with the standard **logit-processor interface** in modern inference engines. Popular mechanisms such as top-$k$/top-$p$ filtering and repetition penalties in vLLM are all implemented in logit space (see vLLM’s sampler implementation [6]). Logit-level fusion integrates directly into this interface, making ThinkMerge plug-and-play in vLLM and the upcoming vLLM+FlexAttention stack.
>
> Beyond the aggregation rule itself, our work provides a systematic exploration of "Diverse Thinking, Answering as One" for challenging reasoning and open-ended tasks. The approach is particularly effective for open-domain code generation (LiveCodeBench) and the **newly added deep-research agent benchmarks:** BrowseComp-en, BrowseComp-zh, GAIA, and XBench-DeepSearch ($\\textrm{\\color{blue}Section 5.4}$ in paper revision).
>
>
> ---
>
> ***W2: The proposed fusion is naive, and can potentially fail in scenarios where the reasoning traces diverge too much.***
>
> We agree that understanding how ThinkMerge behaves when reasoning traces strongly diverge is important. Our experiments are designed to explicitly study this.
>
> (1) We study merging up to $K{=}8$ traces using a relatively **high sampling temperature** (up to 0.7, as described in Appendix A.1). Under such high temperatures, the 8 traces are indeed diverse.
>
> (2) We further ablate the **sampling temperature** (see $\\textrm{\\color{blue}Table 3}$). We observe that, for the same $K$, higher temperatures (more diverse thinking) generally lead to *better* performance than a low temperature of 0.3.
>
> These results indicate that, under commonly used high-temperature settings where traces explore different strategies over relatively large $K$, logit-level ThinkMerge does **not collapse** and in fact benefits from diversity.
>
> While extremely cases with completely contradictory strategies may still exist, our current experiments already cover a practically relevant high-diversity regime. We will make this discussion more explicit in the revised version to better highlight the limitations and robustness properties of ThinkMerge.
>
> ---
>
> ***W3: Token probability ensembling literature needs to be covered more in Related section [2,3,4,5].***
>
> Thank you again for pointing out these related works. We have included [2–5] in our revised Related Work section.

---

> ### Author Response · Authors · 2025-11-21
> **Rebuttal by Authors [2/2]**
>
> ---
>
> ***W4&Q2: While the paper emphasizes open domain tasks, it's experimentally validated only with a single benchmark, LiveCodeBench. Including more benchmarks on open-domain reasoning***
>
> Thank you for raising this point. We address it by significantly expanding our evaluation on open-ended tasks. In the revised version, we add a new subsection ( $\\textrm{\\color{blue}Section 5.4}$) that evaluates ThinkMerge in deepresearch agent settings.
>
> Concretely, we integrate ThinkMerge into the Tongyi-WebSailor web-browsing agent and test on four challenging open-domain benchmarks: **BrowseComp-en**, **BrowseComp-zh** (hard multi-hop fact-finding in English/Chinese), **GAIA** (real-world multi-step tasks with tool use), and **XbenchDeepSearch** (expert-level deep web queries). These tasks require free-form outputs (paragraph-style answers or executed code) and therefore cannot be handled by simple majority voting on final strings.
>
> We keep the agent’s tool-use policy fixed and apply ThinkMerge only at the **final answer generation** stage, comparing a single-run agent to ThinkMerge with $N{=}2,4,8$ independent trajectories. The new results show consistent gains for sufficiently strong agents (7B and 32B), demonstrating that ThinkMerge remains effective beyond LiveCodeBench and supports the paper’s emphasis on open-ended scenarios.
>
> ---
>
> ***Q1: This work can be a good empirical contribution but would require some more work, especially around analysing which scenarios benefit from fusion approach and investigating it's limitations and failure modes.***
>
> We are glad that your assessment that this work can be a good empirical contribution with some additional analysis. In the revision, we have taken steps in this direction along two axes:
>
> (1) **Broader scenarios.** As discussed in our response to W4, we extend our experiments beyond LiveCodeBench to four challenging open-domain web benchmarks with an agentic setup (BrowseComp-en/zh, GAIA, XbenchDeepSearch). These experiments show when ThinkMerge provides clear benefits over a single-run agent and simple majority voting, especially for stronger base agents.
>
> (2) **When fusion helps vs. hurts.** In $\\textrm{\\color{blue}Section 5.5}$, we add a dedicated comparison of Merge Logit vs. Merge Probability for reasoning models. We find that probability-level merging can perform even worse than majority voting on AIME’25, while logit-level merging provides consistent improvements. This analysis directly addresses which fusion schemes are suitable for reasoning, and which ones can be detrimental.
>
> ---
>
> ***References:***
>
> [1]: Product of Experts
>
> [2]: M-Ped: Multi-Prompt Ensemble Decoding for Large Language Models
>
> [3]: Bridging the Gap between Different Vocabularies for LLM Ensemble
>
> [4]: Token-level Ensembling of Models with Different Vocabularies
>
> [5]: Llm-blender: Ensembling large language models with pairwise ranking and generative fusion
>
> [6]: vLLM: https://github.com/vllm-project/vllm/blob/6d8d0a24c02bfd84d46b3016b865a44f048ae84b/vllm/model_executor/layers/sampler.py#L266

---

### Official Review · Reviewer_f3xd · 2025-11-12

**Soundness:** 4
**Presentation:** 4
**Contribution:** 3
**Rating:** 8
**Confidence:** 4

**Summary:**

Authors present ThinkMerge which averages logits over multiple parallel reasoning traces to enable majority voting over open-ended QA. They outperform baselines on AIME, GPQA, and even LiveCodeBench.

First, they demonstrate that Pass@N indeed goes up for all models on all tasks, both open- and close-ended. This serves as a basis for the assumption that majority voting should benefit performance. Further, they find that these gains do come from the model gaining “capability” on difficult questions as inference scale grows.

Their approach follows a map-reduce paradigm. First, they sample K CoTs and then left-pad them so that the end of the CoT (the `</think>` token) in each seq is aligned. To do this they autoregressively have each CoT effectively “vote” token-by-token by averaging the pre-softmax logits that come from each chain.

They considered a few other alternatives before landing on this approach. Across most tested models, their method outperforms simple majority voting when the chains are allowed to fully expand, rather than only using the shortest K, for AIME and GPQA. Additionally, on the open-ended livecodebench, their method consistently outperforms the baseline, and this shortest K early stopping approach still delivers gains. This is useful, as it means that potentially the method can be made more efficient for open-ended tasks.

**Strengths:**

Clear implementation of a straightforward approach to improve reasoning model performance.

Method delivers a clear improvement

Well-described and easy to followN/A I think this is a solid paper.

**Weaknesses:**

No discussion of closed models here. I’m curious, how well does this close the gap between the open and closed models?

**Questions:**

I think this is a pretty solid paper. What do you think about the weakness suggested?

---

> ### Author Response · Authors · 2025-11-21
> **Rebuttal by Authors**
>
> We appreciate the reviewer's encouraging feedback! We are glad that our method is seen as a straightforward approach with a "clear improvement" in reasoning model performance. We address the remaining questions and concerns below.
>
> ---
>
> ***W1: No discussion of closed models here. I’m curious, how well does this close the gap between the open and closed models?***
>
> Thank you for raising this point. Conceptually, ThinkMerge is model-agnostic: it only requires **(a)** the ability to separate thinking (e.g., ```<think>…</think>```), and **(b)** access to the answer-time logits. Once these interfaces are available, the same test-time scaling strategy could be used with closed models as well, and the relative gains we observe on open models should transfer in a largely orthogonal way. In that sense, our work is best viewed as proposing a general test-time compute scaling method for reasoning models.
>
>
> ---
>
> ***W2: I am not sure about the novelty. I would like to see how the authors respond to the concerns raised by other revs.***
>
> We appreciate your willingness to see how we address the novelty concerns. In the revision, we have made several concrete changes in direct response to the other reviewers’ comments:
>
> **Clarifying what is new.** As discussed in our replies to Reviewers CwCp and SGtm, our contribution is in how we instantiate and analyze fusion over parallel reasoning traces in the modern think–then–answer setting. Prior ensemble-decoding work typically operates either at the probability level or at the solution level; we show that probability-level merging can actually underperform majority voting on challenging reasoning tasks (newly added $\\textrm{\\color{blue}Section 5.5}$), whereas logit-level merging yields consistent gains and integrates cleanly into existing inference stacks.
>
> **Plug-and-play implementation.** We provide a logit-processor-style implementation that is directly compatible with widely used inference frameworks (e.g., vLLM-style samplers with top-k/top-p, repetition penalties, and efficient attention backends). This makes ThinkMerge immediately deployable, which we believe is an important practical aspect of the contribution.
>
> **New analysis and ablations.** In the revised $\\textrm{\\color{blue}Section 5.5}$, we add a dedicated Merge Logit vs. Merge Probability comparison on AIME’25, showing that simply reusing probability-level fusion from prior works[1,2,3] (as suggested by Reviewer CwCp) can make reasoning models worse than majority voting, while our logit-space variant consistently improves performance.
>
> **Broader and more open-ended evaluation.** To better substantiate the practical impact, we substantially expand the open-ended experiments beyond LiveCodeBench: the revised paper adds a new subsection on *Open-ended DeepResearch agents*, where we plug ThinkMerge into the Tongyi-WebSailor web-browsing agent and evaluate on four challenging deepresearch agent benchmarks (BrowseComp-en, BrowseComp-zh, GAIA, XBenchDeepSearch). These settings involve tool use, long-horizon reasoning, and free-form answers, where solution-level majority voting is ill-defined. In this regime, ThinkMerge provides consistent gains for sufficiently strong agents (7B and 32B).
>
> ---
>
> **References:**
>
> [1]: M-Ped: Multi-Prompt Ensemble Decoding for Large Language Models - Guo et.al, CORR 2024
>
> [2]: Bridging the Gap between Different Vocabularies for LLM Ensemble - NAACL 2024
>
> [3]: Token-level Ensembling of Models with Different Vocabularies - Rachel Wicks et.al (arxiv)

---

### Author Response · Authors · 2025-12-03
**Summary of Rebuttal Revisions and General Response**

Dear Reviewers and Area Chairs,

Thank you for taking the time to review our paper and for the thoughtful, detailed feedback. We are glad that two reviewers (f3xd, bs9g) found our test-time scaling approach to be a simple and practical way to improve reasoning performance, and that Reviewer f3xd supported the work with an overall score of 8. We also appreciate that Reviewer CwCp, despite a marginally-below-threshold score, explicitly noted that this work "*can be a good empirical contribution*" provided that we more carefully analyze when fusion helps and where its limitations and failure modes lie.

Motivated by these comments, our revision focuses on three fronts: (i) broadening the **open-ended and agentic evaluation** (in web-based DeepResearch settings), (ii) providing a **deeper analysis of fusion schemes and failure modes** (e.g., logit vs probability merging, the effect of K and sampling temperature), and (iii) **clarifying scope, novelty, and limitations**, including the connection to prior ensemble-decoding and map–reduce style methods.

Below we summarize, for each reviewer, the main concerns and what we addressed in the rebuttal and revision:

| Reviewer | Main concerns (very brief) | What we did in rebuttal |
| :--- | :--- | :--- |
| **f3xd** | No discussion of closed models; unclear novelty vs prior ensemble work. | Clarified that **ThinkMerge is model-agnostic** and directly applicable to closed models once think/answer separation and answer-time logits are exposed; highlighted our **logit-level fusion** and added a **Merge-Logit vs Merge-Probability study on AIME’25 (Sec. 5.5)** showing that, for modern reasoning models, probability fusion can underperform majority voting while logit fusion consistently helps; and **expanded open-ended evaluation** with DeepResearch agents beyond LiveCodeBench.|
| **CwCp** | Limited innovation/novelty; naive fusion may fail when traces diverge; incomplete coverage of token-ensembling work; open-domain evaluation too narrow; needs clearer analysis of when fusion helps vs hurts. | Expanded **Related Work (Sec. 2)** to cover M-Ped and other token-level ensembling methods and clarified that prior work ensembles **probabilities**, whereas ThinkMerge ensembles **logits** compatible with vLLM logit processors; added a dedicated **Merge-Logit vs Merge-Probability ablation on AIME’25 (Sec. 5.5)** and analysis under **higher temperatures and larger K**; and broadened open-domain evaluation with **four DeepResearch benchmarks** using Tongyi-WebSailor agents (Sec. 4.3) to show when fusion helps and when it can hurt.|
| **bs9g** | Dependence on clean think/answer delimiters; sometimes matches or loses to majority voting on AIME/GPQA; concern that pure logit averaging can drown a minority-but-correct trace; gains saturate or regress as K increases. | Clarified that ThinkMerge is explicitly targeted at the now-standard **think–then–answer paradigm**; highlighted the **Trimming (De-Repeat Suffix)** heuristic and showed in **Tables 1–2** that ThinkMerge is reasonably robust to noisy or rambling think segments, with only modest additional gains from trimming; repositioned the method’s **main value on open-ended tasks** where majority voting is ill-defined and supported this with new DeepResearch experiments where ThinkMerge gives consistent improvements; acknowledged that confidence-weighted variants could further help minority-correct traces and framed them as future work; and analyzed **K-scaling behavior**, noting that most gains are already captured at small–moderate K (e.g., K=4), so saturation beyond that is a practical trade-off rather than a core failure.|
| **SGtm** | Lack of novelty; evaluation not truly open-ended; method seen as essentially majority voting / generic map–reduce; no significant gains on genuinely open-ended tasks. | For novelty, clarified that ThinkMerge is **not majority voting**, but a **logit-space fusion scheme** instantiated for modern think–then–answer reasoning and distinct from GraphRAG-style map–reduce; refined related work to explicitly discuss **GraphRAG and ensemble methods** and positioned ThinkMerge as a **simple, scalable, plug-and-play recipe with off-the-shelf implementation**. For open-ended evaluation, stressed that LiveCodeBench is already genuinely open-ended (execution-based correctness) with **+7.58 / +8.28** pass@1 gains on its hard subset, and **substantially expanded experiments** with DeepResearch benchmarks (**GAIA, XBenchDeepSearch, BrowseComp-EN/ZH**) where WebSailor-7B/32B obtain **5–10 point** absolute gains.|

We recognize that this year’s ICLR review process is highly competitive, and we are sincerely grateful that you invested time in reading our detailed rebuttal and the extended experimental results. We remain confident that **ThinkMerge** as a plug-and-play strategy for reasoning models, can provide a practical and broadly applicable way for building stronger open-ended reasoning and agentic systems!

---

### Meta-Review · Area_Chair_fiFF · 2025-12-06

**Summary:**

- Reviewer f3xd (8): No discussion of closed models; unclear novelty vs prior ensemble work.
- Reviewer CwCp (4): Limited innovation/novelty; naive fusion may fail when traces diverge; incomplete coverage of token-ensembling work; open-domain evaluation too narrow; needs clearer analysis of when fusion helps vs hurts.
- Reviewer bs9g (6): Dependence on clean think/answer delimiters; sometimes matches or loses to majority voting on AIME/GPQA; concern that pure logit averaging can drown a minority-but-correct trace; gains saturate or regress as K increases.
- Reviewer SGtm (2): Lack of novelty; evaluation not truly open-ended; method seen as essentially majority voting / generic map–reduce; no significant gains on genuinely open-ended tasks.

**Reviewer Concerns:**

All the reviewer concerns listed above were addressed by the rebuttal.

**Reviewer Scores:**

As the concerns are well addressed by the rebuttal, I guess:
- Reviewer f3xd will keep the positive score of 8.
- Reviewer CwCp will raise the score from 4 to 6.
- Reviewer bs9g will keep the positive score of 6.
- Reviewer SGtm will raise the score from 2 to 4.

---

### Decision · Program_Chairs · 2026-01-26

Accept (Poster)